# The Relationship between Estrogen and Nitric Oxide in the Prevention of Cardiac and Vascular Anomalies in the Developing Zebrafish (*Danio Rerio*)

**DOI:** 10.3390/brainsci6040051

**Published:** 2016-10-26

**Authors:** Benjamin G. Sykes, Peter M. Van Steyn, Jonathan D. Vignali, John Winalski, Julie Lozier, Wade E. Bell, James E. Turner

**Affiliations:** Department of Biology, Center for Molecular, Cellular, and Biological Chemistry, Virginia Military Institute, Lexington, VA 24450, USA; sykesbg@vmi.edu (B.G.S.); vansteynpm@mail.vmi.edu (P.M.V.S.); vignalijd10@mail.vmi.edu (J.D.V.); winalskijs16@mail.vmi.edu (J.W.); lozierja@vmi.edu (J.L.); bellwe@vmi.edu (W.E.B.)

**Keywords:** embryonic zebrafish, nitric oxide, estrogen, heart arrhythmias, blood vessel development

## Abstract

It has been known that both estrogen (E2) and nitric oxide (NO) are critical for proper cardiovascular system (CVS) function. It has also been demonstrated that E2 acts as an upstream effector in the nitric oxide (NO) pathway. Results from this study indicate that the use of a nitric oxide synthase (NOS) inhibitor (NOSI) which targets specifically neuronal NOS (nNOS or NOS1), proadifen hydrochloride, caused a significant depression of fish heart rates (HR) accompanied by increased arrhythmic behavior. However, none of these phenotypes were evident with either the inhibition of endothelial NOS (eNOS) or inducible NOS (iNOS) isoforms. These cardiac arrhythmias could also be mimicked by inhibition of E2 synthesis with the aromatase inhibitor (AI), 4-OH-A, in a manner similar to that of nNOSI. In both scenarios, by using an NO donor (DETA-NO) in either NO + nNOSI or E2 + AI co-treatments, fish could be significantly rescued from decreased HR and increased arrhythmias. However, the addition of an NOS inhibitor (L-NAME) to the E2 + AI co-treatment fish prevented the rescue of low heart rates and arrhythmias, which strongly implicates the NO pathway as a downstream E2 targeted molecule for the maintenance of healthy cardiomyocyte contractile conditions in the developing zebrafish. Cardiac arrhythmias could be mimicked by the S-nitrosylation pathway inhibitor DTT (1,4-dithiothreitol) but not by ODQ (1H-[1–3]oxadiazolo[4,3-a]quinoxalin-1-one), the inhibitor of the NO receptor molecule sGC in the cGMP-dependent pathway. In both the nNOSI and AI-induced arrhythmic conditions, 100% of the fish expressed the phenotype, but could be rapidly rescued with maximum survival by a washout with dantrolene, a ryanodine Ca^2+^ channel receptor blocker, compared to the time it took for rescue using a control salt solution. In addition, of the three NOS isoforms, eNOS was the one most implicated in the maintenance of an intact developing fish vascular system. In conclusion, results from this study have shown that nNOS is the prominent isoform that is responsible, in part, for maintaining normal heart rates and prevention of arrhythmias in the developing zebrafish heart failure model. These phenomena are related to the upstream stimulatory regulation by E2. On the other hand, eNOS has a minimal effect and iNOS has little to no influence on this phenomenon. Data also suggests that nNOS acts on the zebrafish cardiomyocytes through the S-nitrosylation pathway to influence the SR ryanidine Ca^2+^ channels in the excitation-coupling phenomena. In contrast, eNOS is the prominent isoform that influences blood vessel development in this model.

## 1. Introduction

It has been well established through many in vivo and in vitro studies that estrogen (E2) is critical for normal development, maintenance, and function of the cardiovascular (CVS) system [1]. E2 as a steroid hormone acts in development as a signaling molecule to affect differentiation by its ability to modulate cell death and development, among other similar actions [1]. There is a vast and complex response system throughout cardiac, blood vessel, and smooth muscle tissues all of which have E2 receptors (ERs). E2 is synthesized from androgens by the enzyme aromatase in both male and female species [1]. A key area of study for applications in medicine for E2 is in therapies for postmenopausal women. Women generally have a lower risk for cardiovascular disease than men. This changes after menopause, and therefore E2 is implicated to be a CVS protectant. Past attempts at E2 replacement therapies, such as the one done by the Women’s Health Initiative in the 90s, was met with unexpected results which reported increased risk of CVS disease and stroke [1]. Therefore, more research is necessary to understand the effects of E2 and its actions on the mature and developing body to better understand possible medicinal applications and implications.

Part of the mechanism by which E2 exerts control over the CVS is through its influence on the expression of nitric oxide synthase (NOS), the enzyme that produces nitric oxide (NO). NO is a gas that acts as a signaling molecule itself in several pathways and genomic mechanisms [2,4]. By virtue of its gaseous state, NO can diffuse across cellular membranes without the aid of membrane bound transport proteins and can interact directly with its end targets either in the cell in which it was synthesized or in surrounding cells. In turn, its actions are precisely controlled due to its very short half-life [3,5]. At high concentrations, NO acts as a free radical in some situations or binds to superoxide anion (O_2_**^.^**), causing pathophysiological effects that act to harm the body [6]. Conversely, at lower levels NO has been shown to exhibit protective effects within the CVS [6]. In addition, NO acts to regulate cardiac adaption on a spatio-temporal dimension by affecting the heart through the Frank–Starling mechanism, cardiac contractility at the biochemical level, and altering gene expression leading to long term effects [7]. There are four isoforms of NOS that synthesize NO from L-arginine: neuronal (nNOS or NOS1), inducible (iNOS or NOS2), endothelial (eNOS or NOS3), and mitochondrial (mtNOS). E2 is implicated in the expression of nNOS and eNOS in the CVS [1]. nNOS is expressed throughout the body in various tissues, but most notably in nervous tissues and cardiomyocytes and plays physiological roles in neurogenesis, learning, memory, and regulation of blood pressure [6]. iNOS does not have much support from the literature to suggest it has a significant role in the function of the CVS [8].

eNOS acts as a potent regulator of the cardiovascular system, most notably in the vasculature as a potent vasodilator, but also in cardiomyocytes. eNOS derived NO acts to increase the amount of sGC and thereby increasing cellular cGMP which leads to vasodilation [6]. All four isoforms are found throughout the cardiovascular system [1]. Normal physiological function of the heart and vascular tissues is balanced by NO expressed by all isoforms that work together to regulate contractility and peripheral and central nervous system control of the heart itself [5,9,10].

NO acting at the cellular level interacts with either soluble guanylyl cyclase (sGC) to produce cyclic GMP (cGMP) or causes *S*-nitrosylation of cysteine residues, as well as interacting with other pathways in the cell [7,11]. sGC in cardiomyocytes can be activated in an autocrine manner by NO produced in the myocytes, or in a paracrine manner by NO produced in non-myocyte cells surrounding the cardiomyocyte such as endothelial cells, fibroblasts, smooth muscles, and neurons [5]. sGC’s heme group has a higher binding affinity for NO than for molecular O_2_ in the cytoplasm, which is critical for the NO/cGMP dependent-pathway since NO is present in much lower concentrations in the cell than O_2_ [12]. Once cGMP is produced in this pathway, it acts to modulate phosphodiesterases, ion-gated channels, or cGMP-dependent protein kinases (PKG), each of which continues to act physiologically in the cardiovascular system and throughout the body in vasodilation, platelet aggregation, and neurotransmission [12].

The other major pathway activated by NO is that of *S-*nitrosylation, which is the posttranscriptional modification of cysteine residues by covalently bonding of NO replacing hydrogen on the sulfur group and changing the configuration of a protein [2,11]. These proteins then act as second messengers and signal effectors [2]. In the vasculature, *S*-nitrosylation has been implicated in downstream NO vasodilatory signaling, as well as the onset of atherosclerosis [2]. In cardiomyocytes *S*-nitrosylation seems to have protective functions against the development of arrhythmias by way of the nNOS derived NO pathway [2]. This action involves the inhibition of L-type Ca^2+^ channels by *S*-nitrosylation. *S-*nitrosylation has also been shown to act on ryanodine receptors of the sarcoplasmic reticulum of cardiomyocytes to regulate intracellular Ca^2+^ and thus heart contractility [2]. All these functions help show the importance of *S-*nitrosylation in the function of the CVS and the body as a whole.

Zebrafish are commonly accepted model organisms for use in research labs. They exhibit a very quick developmental life cycle, and can reproduce in large numbers. They are also relatively translucent, making study of internal organs and blood vessels easily visible. They also use many of the same genetic and biochemical pathways that are conserved in mammals and other higher organisms. The zebrafish also exhibits a robust NO response system. Specifically, nNOS is expressed early in zebrafish development at 16 to 19 h post fertilization (hpf). Zebrafish are also responsive to nNOS inhibitors (nNOSI), as well as many of the dependent and independent pathway agonists and antagonists [13]. The nNOS isoform is also expressed in the developing zebrafish medulla, a region of the brain that is known to control heart rate [14]. Also, blood vessel development occurs very quickly in the zebrafish and blood begins to circulate in the blood vessels after 24 hpf [15]. Overall, nNOS plays an important role in the zebrafish cardiovascular system [11].

Previous work in our lab has described the “listless” model’ which is initiated by E2 and NO deprivation [13,16,17]. Specifically, both E2 and NO deficiencies cause reduced heart rates and increased arrhythmias, as well as initiating severe sensory-motor deficits. From these earlier observations, it is hypothesized that it is the nNOS isoform that is more effective in regulating heart rate and preventing arrhythmias in the developing zebrafish CVS. In addition, it is also hypothesized that eNOS plays the most prominent role in vascular bed development and health in the developing zebrafish. Results from this study validate these hypotheses. Specifically, nNOS was found to be the prominent isoform regulator of heart health preventing decreased heart rates with accompanying arrhythmias, and is a downstream effector of E2 stimulation. In contrast, eNOS is the isoform which has the most influence on fish blood vessel development.

In summary, this study will help further our knowledge of how the developing heart and blood vessels utilize the various NO signaling pathways in both health and disease. These findings will also aid in medical research and improve our understanding of the use of more novel pharmacological approaches in heart patients suffering for congestive heart failure (CHF) and arrhythmic symptoms.

## 2. Methods

### 2.1. Fish Preparation

Upon arrival, fish were placed in an autoclaved embryo rearing salt solution (ERS) composed of 0.04 g of CaCL_2_, 0.163 g of MgSO_4_, 1.0 g of NaCl, and 0.03 g of KCL all in 1 L of deionized water containing a 0.05% methylene blue solution which serves as an antifunfal agent. All solutions were changed every 24 h and embryos were incubated at 28 °C. All reagents were obtained from Sigma-Aldrich (St Louis, MO, USA) unless noted otherwise, and solutions were made daily before use. A protease concentration of 2 mg/mL (Sigma) was used to dechorionate those embryos treated at 48 h post fertilization (hpf), in order to better expose the fish to the various treatments [18]. Fish treated at 4–6 days post fertilization (dpf) were allowed to hatch on their own. All procedures were in accordance with NIH guidelines for the care and treatment of animals.

Several fish strains were used in the current study. Wild-type embryos (strain AB) were obtained from either the Zebrafish International Resource Center (University of Oregon, Eugene), the breeding facilities at Virginia Military Institute, or Roanoke College. A second fish used was the compound *roy;nacre* double homozygous mutant, which is named *casper* and shows the effect of combined melanocyte and iridophore loss in which the body of the embryonic and adult fish is largely transparent [19]. In addition, two transgenic models were used to study either heart or blood vessel dynamics. The *Tg(fliα:EGFP)^y1* strain [20], was used, which expresses GFP in blood endothelial cells, allowing for live imaging to be done on the developing blood vasculature. A second transgenic line, the *Tg(cmlc2:gCaMP)* fish, incorporates gCaMP which acts as a Ca^2+^ sensor because it is voltage sensitive to Ca^2+^ and fluoresces in its presence , such as during cardiomyocyte contraction [18]. This line has been used to map Ca^2+^ flux in other experiments and is being used in the current study to help determine Ca^2+^ flux relating to heart rate and its role in arrhythmias under various treatment conditions [21].

### 2.2. Reagent Preparations

All NO-related reagents for treating zebrafish were tested in a dose–response paradigm to ensure optimal results and proper survival. Based on the literature, baseline target concentrations were identified. Nitro-l-arginine methyl ester hydrochloride (L-NAME-hydrochloride, Sigma) was used as a general NOS inhibitor (gNOSI) and tested at 10, 15, 25, and 30 mM concentrations which were prepared in ERS. The optimal concentration for L-NAME solution was found to be 15 mM. gNOSI inhibits all three isoforms of NOS by acting as an L-arginine analog [22]. Proadifen hydrochloride (Sigma) was used as a selective nNOS inhibitor (nNOSI). With ERS as the diluent, fish were tested at 10 μM, 30 μM, and 50 μM. The 50 μM concentration provided optimal results in its ability to lower heart rates and increase arrhythmias. *S*-methylisothiourea sulfate (SMT, Sigma) was used as a selective iNOS inhibitor (iNOSI). With ERS as the diluent, it was tested at 100 μM, 10 μM, and 1 μM. It was found that 10 μM provided optimal results. Diphenyleneiodonium chloride (Sigma) was used as a selective eNOS inhibitor (eNOSI). It was dissolved in 0.1% DMSO solution and ERS added to dilute to 100 μM, 10 μM, 1 μM, 100 nM, 10 nM, and 1 nM, which was further refined to 3 μM, 5 μM, and 7 μM. Of these concentrations, 5 μM was the most effective treatment.

The AI 4-hydroxy androstenedione (4-OH-A, Sigma) was used at a 50 μM concentration, and estrogen (17β-Estradiol, Sigma) at 10 nM as established previously [16]. Both were dissolved in 0.05% ethanol and diluted to the appropriate concentration with and egg rearing solution (ERS). The control group consisted of the ERS salt solution which contained 0.05% ethanol.

1H-[1–3]Oxadiazolo[4,3-a]quinoxalin-1-one (ODQ, Sigma) was used as a soluble guanylyl cyclase (sGC) inhibitor which compromises the NO-cGMP-dependent pathway by reducing cGMP production. It was dissolved into a 0.1% DMSO solution then diluted with ERS to a working concentration of 30 μM for application. In addition, DTT (dithiothreitol, Sigma) was used as an inhibitor of the NO-cGMP-independent pathway which prevents S-nitrosylation events.

Diethylenetriamine/nitric oxide adduct (DETA-NO, Sigma) was used to provide a slow extended release of exogenous NO as a co-treatment with some of the inhibitors used in the experiments in an effort to show that NO inhibition mediated symptoms exhibited by fish can be rescued. It was dissolved into ERS resulting in working concentrations of 400 μM to 50 μM with 50 μM providing the best results.

Dantrolene sodium salt (Sigma), a ryanodine Ca^2+^ channel receptor blocker, was solubilized in 0.1% DMSO for a stock solution of 1 mM. The working solution to achieve maximal results in our experiments was 10 µM.

### 2.3. Heart Rate, Arrhythmia and Ca^2+^ Flux Analyses

Brightfield microscopy was performed using a Nikon camera and NIH ImageJ program to observe heart rates and overall health of the zebrafish. Fish were placed into 30 mm petri dishes and if necessary were immersed in minimal medium to prevent fish from moving, which was done for short periods of time to facilitate imaging. Heart rates were counted for 30 s and then multiplied by two to find the heart rate in beats per minute (bpm). At least six fish per treatment group were analyzed. While fish were onscreen, arrhythmic heart beats were also observed and scored according to the NIH Heart, Blood and Lung Institute, as any inconsistent heart rate, either too slow or fast, as well as irregular in nature. In particular, heart rates below 120 bpm (160 bpm is a normal rate) were classified as arrhythmic. Fish mortality was confirmed when heart function had ceased. Any abnormal heart structural observations were photographed. The *Tg(cmlc2:gCaMP)* line of transgenic fish expresses a voltage-sensitive fluorescent Ca^2+^ sensor in cardiomyosites [18]. Therefore, repetitive fluorescent waves were visualized and/or recorded as a systolic Ca^2+^ release spreading from the atrium via the AV-junction to the ventricles whose peaks and intervals were analyzed for arrhythmic behavior [21]. Specifically, GFP tagged transgenic fish were anesthetized using a tricaine solution just sufficient to provide immobilization and were placed in a glass bottom culture dish. The fish were imaged using a Nikon Eclipse Ti inverted microscope with a filter set appropriate for the GFP tag. The ventricle of the heart was identified and selected in the Nikon Elements software as a region of interest. Data was collected for 30 s on each animal. Graphical data representing fluctuating levels of calcium ions were analyzed for intensity and periodicity.

### 2.4. Blood Vessel (Angiogenesis) Analysis

Confocal microscopy was performed at Washington and Lee University on an Olympus Confocal microscope. *Tg(fliα:EGFP)^y1* fish were used for blood vessel imaging [19]. Fish were anesthetized using Tricaine solution to ensure fish would not move during imaging and to ensure that the live specimens could be properly photographed. Specimens were placed either in 30 mm dishes with glass bottom cover slips or plastic dishes and prepared for imaging by placing them on their sides to get an ideal image of the vasculature in the tail region. A series of z-stack images of the vasculature in the tail of the zebrafish was compiled and then analyzed for malformations and other irregularities in blood vessel development using NIH ImageJ software and Microsoft Live Photo Viewer. The four main vessels that were analyzed were caudal artery (CA), vertebral artery (VA), intersegmental vessels (SE), and the dorsal longitudinal anastomotic vessel (DLAV). Specifically, the caudal artery (CA) diameter, length between the end of the dorsal longitudinal anastomotic vessel (DLAV) and last intersegmental vessel (SE), and vertebral artery (VA), and the quality of the SE were all documented. The quality of the SE vessel was assessed by looking for three different mistakes in vessel formation, (1) the number of SE bifurcations; (2) the number of VA/SE misconnections; and (3) the number of gaps in the VA, were documented. A SE bifurcation was recorded any time the SE artery and vein of one myotome bifurcate moved away from each other and connected to the DLAV in the improper locations. CA diameter was measured at 875 µm from the caudal end of the vessel bed in order to establish a constant standard reference point. To determine the state of DLAV development or degeneration, the gap between where the DLAV ceased to exist continuously and where the last SE was located was measured. All VA/SE misconnections were recorded any time the VA connected with another SE in an anterior or posterior region instead of connecting centrally, which is the proper developmental location for the connection to occur. Whenever a gap appeared in the VA between SE it was documented as a skipped VA connection.

### 2.5. Statistical Analysis

Thirty zebrafish embryos were used in each experimental and a control group within the various experimental paradigms described above. Each experimental study was repeated a minimum of three times for statistical interpretation. Statistical analysis was performed using SigmaStat 3.5 software, Microsoft Excel, and VasserStats analysis website. Heart rate and blood vessel data was entered into either an ANOVA one-way paired t-test or ANOVA Single Factor tests. For the ANOVA analysis, a Holm–Sidak post hoc method was also run to determine significant differences between the various treatment groups. Percent survival and arrhythmic heart rate data were analyzed using either a *z*-test for two population proportions or for multiple proportions using a chi-square contingency table test, followed by a Marascuilo’s post hoc analysis. All experiments were repeated in triplicate.

## 3. Results

### 3.1. E2 and NO Are Linked in the Maintenance of Heart Rate

Effects of various treatments on fish heart rates (HRs) in beats/minute (bpm) after 4 days of treatment beginning at 48 h post fertilization (hpf) are shown in Figure 1. Specifically, both AI and gNOSI significantly reduce HRs by approximately 50% and 25% respectively (*p* < 0.001, Figure 1A). Both sets of depressed HRs can be significantly rescued (*p* < 0.001) with either E2 or DETA-NO co-treatments respectively (Figure 1B,C). On the other hand, gNOSI can prevent the rescue of HRs caused by E2 replacement therapy (Figure 1C). Specifically, while gNOSI is added to the AI + E2 co-treatment fish the rescue of HRs is significantly eliminated (*p* < 0.001) and returns to the original AI-mediated depressed level (compare Figure 1A,C). Figure 2C further shows the linked relationship between E2 and NO HR regulation. Specifically, a significant rescue effect could be elicited in a co-treatment paradigm by combining either nNOSI or AI with DETA as an NO donor (*p* < 0.002 for nNOSI + DETA-NO vs. nNOSI and *p* < 0.001 for AI+DETA-NO vs. AI). These data would indicate that E2 and NO are linked in the maintenance of heart rate with NO as the downstream effector in this relationship.

### 3.2. nNOS Is the Isoform Most Responsible for the Maintenance of HR and Arrhythmia Protection in an Age-Dependent Manner

Among those three NOS isoform inhibitors, only nNOSI significantly decreased the HR (*p* < 0.001; Figure 2A). Both eNOSI and iNOSI HR values were not significantly different from each other or that of controls (*p* > 0.05). The nNOSI effect on HR was dose dependent in fish treated at 2 dpf with analysis at 4 dpf (Figure 2B). Specifically, nNOSI at 30 and 50 μM significantly lowered HR when compared to the 10 µM concentration (*p* < 0.05). However, if fish were treated later at 4–6 dpf, instead of 2 dpf, this dose–response relationship became more pronounced and visualized much sooner at 8 h compared to that at 4 days post-treatment respectively (compare Figure 2B and Figure 3A). In turn, fish survival under these conditions was much more robust at the later development treatment times. Most significantly, the nNOSI-reduced HRs could be brought back to control levels in 100% of the 4–6 dpf treated population by 2.5 days after a washout with the ERS control solution (Figure 3A). These results indicate that fish at later developmental time periods (4–6 compared to 2 dpf) are more responsive to NO deprivation and that nNOS is the principal isoform responsible for this stabilization.

### 3.3. nNOSI and AI Treated Fish HRs Are Arrhythmic and Linked to the NO/cGMP Independent-Pathway Whose Recovery Is Accelerated by Dantrolene Treatment

Effects of nNOSI concentrations on fish heart arrhythmias (HRs below 120 bpm) after treatment beginning at 4–6 dpf are seen in Figure 3. Specifically, there is a significant dose dependent nNOSI relationship related to the induction of heart arrhythmias. In fish treated at 4–6 dpf this dose–response relationship was pronounced and visualized as early as 8 h post treatment compared to similar treatments at 2 dpf which were only evident 2 days later (compare Figure 2B with Figure 3B). Specifically, among these dose–response concentrations there was a significant difference between the 30 and 50 µM nNOSI treatment groups and that of controls (*p* < 0.001). Under these conditions, 50 µM nNOSI elicited the arrhythmic heart phenotype in 100% of the treated population compared to only 10%–15% in fish treated at 2 dpf (Figure 3B,C). AI treatment is also shown to mimic the arrhythmic phenotype in 100% of the fish population treated under the same conditions (Figure 4A). Figure 4B demonstrates that 100 µM DTT treatment, which is an inhibitor of the NO-cGMP-independent pathway and prevents *S*-nitrosylation events, elicits the arrhythmic phenotype in 100% of the fish population after 8 h of treatment of 6 dpf fish (*p* < 0.005). In contrast, no phenotypic expression is evident with a 50 µM ODQ treatment, which blocks the activity of sGC in the NO/sGC/cGMP-dependent pathway, and is no different than that of the ERS controls (*p* > 0.05). These results confirm that both NO and E2 deprivation induced heart arrhythmias that are much more pronounced in the treated populations at 4–6 dpf compared to 2 dpf.

Ca^2+^ wave pattern analysis in the *Tg(cmlc2:gCaMP)* line of transgenic fish after 4 days of treatment beginning at 48 h post fertilization (hpf) confirms the visual recording of the arrhythmic phenotype by demonstrating the effects of nNOSI and AI treatments on selected representative fish hearts compared to controls (Figure 5). The repetitive recording of fluorescent waves represent the systolic Ca^2+^ release spreading from the atrium via the AV-junction to the ventricles whose peaks and intervals were analyzed for arrhythmic behavior [18]. Note the regularity of the spike patterns represented by the peaks and valleys in a control zebrafish heart treated in ERS solution (Figure 5A). For example, this regular interval is repeated at every 0.4 s (see quantitative comparisons in Figure 5D. Note that peaks in a zebrafish heart treated with the aromatase inhibitor (AI) 4-androstene-3,17-dione are farther apart than that of either nNOSI or ERS controls (Figure 5B). Specifically, the time intervals between peaks are measured at a range of between 1.4–1.8 s, which are more than four times longer than ERS controls; however, the spike patterns still remain more regular than their nNOSI treated counterparts (see comparisons between Figure 5C,D below). In the Figure 5C there is a more pronounced arrhythmia behavior exhibited by irregular Ca^2+^ wave patterns in a zebrafish heart treated with 50 μM nNOSI. When compared to either ERS controls or AI treated fish, the interval differences between peaks were observed to be measured over a wide range of between 0.6–1 s and were very irregular in shape and pattern. Also, note that the intensity of the voltage-sensitive fluorescent Ca^2+^ sensor was considerably less in both AI and nNOSI treatment groups when compared to that of the control. Figure 5D quantifies calcium wave values for the three fish seen in Figure 5A–C. Specifically it is shown that the spike pattern is repeated every 0.4 s in the ERS treated control fish which equates to a HR of approximately 150 ± 10 beats/minute which is in agreement with current and published data for control fish [13].). AI heart rates equated to 45 ± 8 bpm compared to 75 ± 10 for that of the nNOSI fish both of which fit the arrhythmic parameters (HR < 120 bpm). Also, note that the Ca^2+^ peak values/min for all three fish matches very closely the actual measured HRs for respective treatment groups throughout the current study (compare with Figure 2C).

Dantrolene, a ryanodine Ca^2+^ channel blocker, washout caused an acceleration of the recovery to normal levels of HRs without arrhythmic behaviors (Figure 6). By 24 min after dantrolene washout, approximately 75% HR recovery was recorded (*p* < 0.001). By 30 min after ERS washout, although approximately 60% of the fish had recovered from the arrhythmic phenotype, they were still significantly behind the 90% recovery for the dantrolene treated fish (*p* < 0.05). By 1 h after washout, both treated groups had returned to normal HR activity.

### 3.4. eNOS Is the Prominent Isoform Involved in Blood Vessel Angiogenesis and Maintenance

The CA diameter is significantly decreased in gNOSI treated fish when compared to that of controls (*p* < 0.001, Figure 7A). Also, note that there was equal significant effectiveness of the three NOSI isoforms similar in reducing CA diameter to that of gNOSI. The number of intersegmental vessel (SE) abnormal bifurcations increased significantly in both gNOSI and eNOSI, but not in nNOSI and iNOSI, compared to ERS controls (Figure 8A; *p* < 0.001). The number of SE misconnections (Figure 8B) increases significantly when treated with gNOSI compared to the three isoforms and control values (*p* < 0.001). Confocal z-stack imagery of the vasculature shows either retarded growth or deterioration, particularly in the dorsal longitudinal anastomotic vessel (DLAV) and vertebral artery (VA) vessels, which appear patchy or absent when compared to control fish (Figure 8C,D). Specifically, significant vertebral artery (VA) misconnections, when treated with either gNOSI or eNOSI, compared to the control values (*p* < 0.001, Figure 8E). In contrast, both nNOSI and iNOSI were ineffective in perpetuating this vessel anomaly (*p* > 0.05). Therefore, at this stage of blood vessel development, eNOS deprivation compromised several aspects of SE and VA vessel development with no apparent engagement from the other two NO isoforms.

## 4. Discussion

Data from the present study suggests that E2 is an upstream regulator of NO-mediated effects on HR and arrhythmias in the developing zebrafish. Specifically, cardiac arrhythmias could be mimicked by inhibition of E2 synthesis with an AI in a manner similar to that of nNOSI. In both scenarios, by using an NO donor (DETA-NO) in either NO + nNOSI or E2 + AI co-treatments, fish could be significantly rescued from decreased HR and increased arrhythmias. However, the addition of an NOS inhibitor (L-NAME) to the E2 + AI co-treatment fish prevented the rescue of low heart rates and arrhythmias, indicating that E2 is an upstream regulator of these events. There is an abundance of molecular, cellular, biochemical, animal model and human patient literature to support the concept that E2 impacts the CVS in significant ways that may involve both genomic and non-genomic mechanisms [1]. Indeed, a robust E2 response system exists within heart, vessel endothelial, and vessel smooth muscle cells that express E2 receptors, as well as the enzyme aromatase, which is responsible for the synthesis of E2 from androgens [1]. Consequently, many prominent CVS physiological effects have also been attributed to the actions of E2 including vasodilation, anti-oxidant properties, decreased post-ischemic inflammation, and anti-artherogenesis effects [23]. As confirmed in the present study, part of the mechanism by which E2 exerts control over the CVS is through its role in regulating the release of NO through increasing the expression of NOS [24].

nNOS is the principal NO source for cardiomyocytes and plays vital roles in all cardiac functions that range from contractility to microcirculation, as well as the principle NO source from the autonomic innervation [4]. Cardiomyocyte contraction is dictated by Ca^2+^ signaling. Ca2^+^ cycling is crucial for excitation-contraction coupling (ECC), which starts when the membrane of the cardiomyocyte depolarizes via voltage-gated Na^+^ channels and causes Ca^2+^ to enter the cytosol through L-type Ca^2+^ channels [2]. Increased cytosolic Ca^2+^ triggers the release of more Ca^2+^ from the SR through ryanodine receptor/Ca^2+^ release channels (RyR2), which then leads to Ca^2+^ binding to troponin C, activating myosin ATPase which leads to the cardiomyocyte, and therefore the heart, contracting. Elimination of Ca^2+^ from the cytosol leads to relaxation of the cardiomyocyte [2]. nNOS also plays a role in CNS control of the heart, separate from its affects within the cardiomyocytes [6]. The NOS isoforms, nNOS and eNOS, both act as regulators of cardiomyocyte contractility through Ca^2+^ cycling by regulation of L-type Ca^2+^ channels and ryanodine receptors (RyR) in the membrane of the SR [25]. These isoforms use similar substrates and cofactors, and depend on Ca^2+^ but exert opposite inotropic effects in cardiomyocytes [6]. nNOS and eNOS have also been shown to be found in specific subcellular locations, giving insight as to their specific function in a cardiomyocyte [26]. Therefore, our findings, that of the three NOS isoforms tested nNOS was the one which most prominently affected the zebrafish HR, fits with this concept. Specifically, evidence points to a role for nNOS in regulating heart contraction, relaxation, and arrhythmic bahavior. There is evidence that nNOS actually attenuates the HR in a number of species but in others may have an opposite effect [4]. Our findings show that nNOSI treatment significantly lowers the developing zebrafish HR, as well as a dramatic increase in the arrhythmic phenotype, similar to E2 deprivation in our previous study [18]. This may have to do with our additional findings that over 90% of the nNOSI treated population also express arrhythmias. Indeed, mice deficient in nNOS demonstrate a diastolic Ca^2+^ leak [27], which creates contractile dysfunction and an arrhythmic state [28,29] much as we see in our model. Also, nNOS has been found to localize on the sarcoplasmic reticulum (SR) in cardiomyocytes [11,25]. nNOS is mainly regulated by Ca^2+^ levels in the cytosol, and is inactive at basal concentrations but activated once levels rise, and is specifically activated by allosteric binding of calmodulin [22]. NO synthesized by NOS has been implicated to increase the open probability of SR Ca^2+^ release channels (the ryanodine receptor isoform, RyR2) [30]. In mammals, NO acts directly by way of *S*-nitrosylation of cysteine residues to increase the open probability by activating RyR2 [2]. This, in turn, enhances contraction of the cardiomyocyte. Increased intracellular levels of calcium lead to increased inotropic activity, which is the force of myocyte contraction, of the cardiomyocyte. Conversely, Wang et al. [30] used nNOS^−^/^−^ knockout mice to show how *S*-nitrosylation levels of RyR2 channels were decreased, leading to decreased open probability, which in turn leads to decreased inotropic activity by way of lower Ca^2+^ sparks [31]. The *S*-nitrosylation pathway was further validated when they treated the knockout mice with *S*-nitroso-*N*-acetylpenicillamine (SNAP), which acts as an NO donor and nitrosylation agent, and observed normalization of cardiomyocyte function. nNOS has been shown to be possibly anti-arrhythmogenic through its inhibition of L-type Ca^2+^ channels by *S*-nitrosylation when it relocates to the sarcolemma during heart failure, as well as regulating β-adrenergic (β-AR) responsiveness and Ca^2+^ entering the cell from the extracellular environment [2]. Arrhythmic heart contractions seem to be intimately related to nNOS expression, as mice that lacked nNOS presented diastolic Ca^2+^ leaking, leading to contractile dysfunction. Regulation of heart action potentials is accomplished by several ion channels that are themselves regulated by S-nitrosylation; an important example of one of these is RyR2 S-nitrosylation by nNOS, which helps regulate intracellular Ca^2+^ concentration [2]. Results from the current study again confirm this hypothesis in that DTT, an S-nitrosylation pathway inhibitor but not ODQ, an NO/sGC/cGMP pathway inhibitor, caused arrhythmias in 100% of the treated embryonic zebrafish population. eNOS is also active in cardiomyocytes using the *S*-nitrosylation pathway to modify L-type Ca^2+^ channels and thus contractility [2]. eNOS may also act in an anti-atherosclerotic manner by preventing leucocytes from adhering to the walls of blood vessels, a key event in the beginning stages of atherosclerosis [6]. eNOS has also been implicated in blood vessel angiogenesis by its mediation of signals that lie downstream of angiogenic factors [6]. Although all isoforms of NOS bind to calmodulin, nNOS and eNOS are more heavily regulated by Ca^2+^ release and calmodulin binding than iNOS [6].

Mechanisms of cardiomyocyte Ca^2+^ handling in zebrafish have not been completely explored; however, a recent study suggests that Ca^2+^ handling in zebrafish ventricular cardiomyocytes may be initiated mainly via either the L-type Ca^2+^ and/or the SR ryanodine channels [32]. Specifically, *S-*nitrosylation has also been shown to act on ryanodine receptors of the sarcoplasmic reticulum of cardiomyocytes to regulate intracellular Ca^2+^ and thus heart contractility [2]. Data from the current study points to the importance of SR ryanodine channels in arrhythmic zebrafish. Specifically, our findings show that washout with a ryanodine receptor blocker, dantrolene, accelerates recovery of fish from the nNOSI induced arrhythmias and cardiac death. The hypothesis here is that dantrolene washout serves to shut off the Ca^2+^ leak faster than the ERS control thus restoring HRs and arrhythmias to normal levels. Also, it would appear from the ODQ and findings that the preferred NO pathway in establishing developing zebrafish cardiac stability is the NO/cGMP independent-pathway. Specifically, the current data demonstrated that DTT treatment, which is an inhibitor of the NO-cGMP-independent pathway by preventing S-nitrosylation events, elicits the arrhythmic phenotype in 100% of the fish population. In contrast, no arrhythmic phenotypic expression is evident with ODQ treatment, which blocks the activity of sGC in the NO/sGC/cGMP-dependent pathway. Other reports have also shown that in cardiomyocytes, *S*-nitrosylation seems to have protective functions against the development of arrhythmias by way of the nNOS derived NO pathway [2].

This study also tested the effects of the three NOS isoforms (eNOS, nNOS, and iNOS) on the integrity and maintenance of the developing zebrafish vascular bed. Angiogenesis is the process of new blood vessel development from the migration and restructuring of endothelial and other necessary cells from pre-existing cells of the vasculature [33]. One of the primary signaling molecules associated with angiogenic stimulation is the vascular endothelial growth factor (VEGF), which, in part, acts through NO to affect vascular development [22]. VEGF activity can be induced by E2, which also induces NO and fibroblast growth factor-2 and its isoforms, making E2 crucial for angiogenesis [34]. The NO-sGC/cGMP dependent-pathway seems to act on mitosis and migratory behavior of endothelial cells, which is critical for the development of new blood vessels [22]. It has been shown that sGC expression coincides with angiogenesis and inhibition coincides with significantly reduced angiogenesis [33].

NO increases endothelial cell growth and migration even in the absence of sGC, but when NO is combined with increased levels of sGC, it has been observed that the angiogenic influence of NO is increased dramatically [35]. However, VEGF has also been shown to act through *S*-nitrosylation to stimulate angiogenesis through the activation of mitogen-activated protein kinase phosphatase 7 (MKP7), which activates c-Jun N-terminal kinase 3 (JNK3), which facilitates endothelial cell migration, a key event in angiogenesis [2]. NO mediates both of these pathways since it activates both sGC and *S*-nitrosylation once it has been stimulated by VEGF [2,16,17]. eNOS has been shown to be active in many different functions of the vascular system, from angiogenesis to anti-atherosclerotic activities, but nNOS also is expressed in low levels in vascular smooth muscle to maintain some vasodilation when eNOS is not active or inhibited or otherwise incapacitated [6].

The current data indicated that eNOS was the isoform most implicated in the maintenance of an intact developing fish vascular system. Specifically, eNOS deprivation resulted in numerous intersegmental vessel misconnections, as well as patchy development or deterioration in the dorsal longitudinal anastomatic vessels and vertebral arteries. These vessels are some of the last to develop in the zebrafish and are thus most sensitive to the various eNOSI manipulations during our study period of 2–4 dpf [13]. In a previous study, we reported similarities in vessel anomalies during development in E2 deprived AI-treated fish, which could be rescued with E2 co-treatment [13]. These comparisons and similarities may be explained by the fact that E2 exerts control over the CVS, in part through the regulation of NOS activity [2,4] either directly or through its stimulatory effects on VEGF activity [34]. Specifically, VEGF activity is induced by E2, which also induces NO and fibroblast growth factor-2 and its isoforms, making E2 crucial for angiogenesis [34]. Specifically, NO has been shown to be one of the primary signaling molecules associated with angeogenic stimulation through VEGF, which, in part, acts through NO to affect vascular development [22]. The NO-sGC/cGMP pathway seems to act on mitosis and migratory behavior of endothelial cells, which is critical for the development of new blood vessels [22]. sGC expression coincides with angiogenesis and its inhibition coincides with inhibited angiogenesis [33]. NO increases endothelial cell growth and migration even in the absence of sGC, but when NO is combined with increased levels of sGC the angiogenic influence of NO is increased dramatically [33]. However, VEGF also acts through *S*-nitrosylation to stimulate angiogenesis through the activation of mitogen-activated protein kinase phosphatase 7 (MKP7), which activates c-Jun N-terminal kinase 3 (JNK3) which facilitates endothelial cell migration, a key event in angiogenesis [2]. NO mediates both of these pathways since it activates both sGC and *S*-nitrosylation once it has been stimulated by VEGF [2,22].

## 5. Conclusions

Results from this study have shown that nNOS is the prominent isoform that is responsible, in part, for maintaining normal heart rates and prevention of arrhythmias in the developing zebrafish heart failure model. These phenomena are related to the upstream stimulatory regulation by E2. On the other hand, eNOS has a minimal effect and iNOS has little to no influence on this phenomenon. Data also suggests that nNOS actions on the zebrafish cardiomyocytes through the S-nitrosylation pathway to influence the SR ryanidine Ca^2+^ channels in the excitation-coupling phenomena. In contrast, eNOS is the prominent isoform that influences blood vessel development in this model. This study’s findings and the development of this model will help further our knowledge of how the developing heart utilizes the various NO signaling pathways in both health and disease. This will also aid in medical research and improve understanding of pharmacological aids, especially in heart patients suffering for CHF and arrhythmic symptoms.

## Figures and Tables

**Figure 1 brainsci-06-00051-f001:**
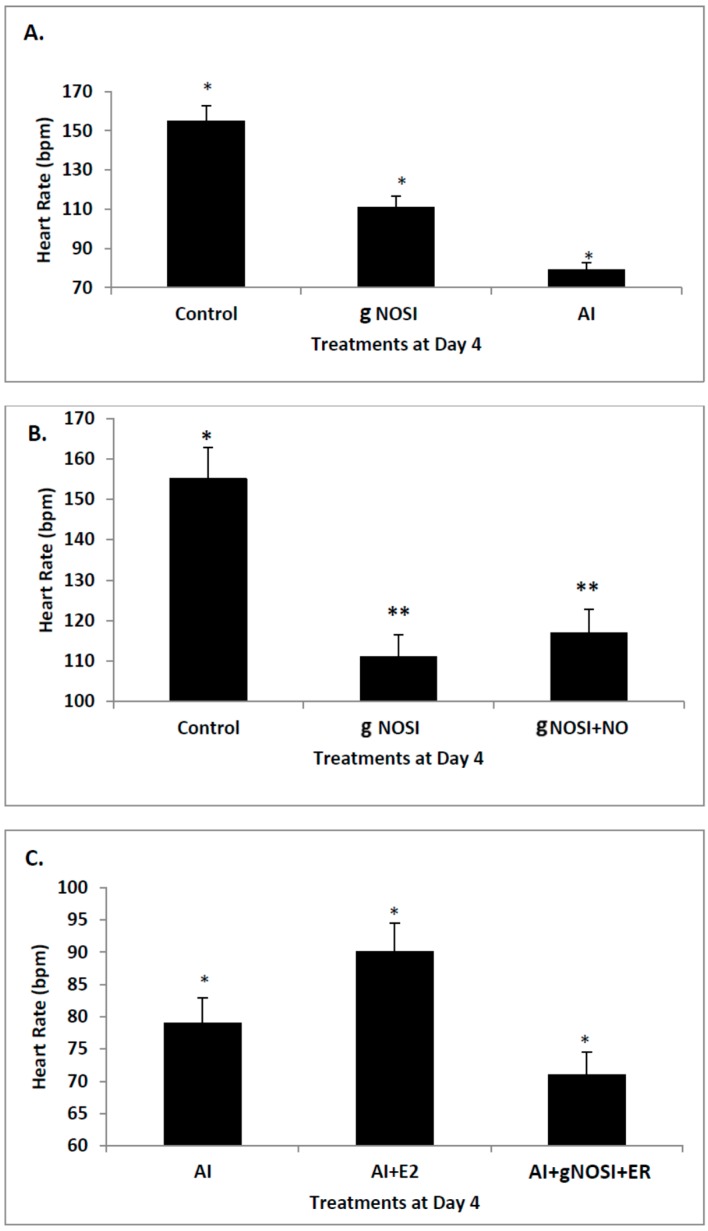
The effects of various treatments on fish heart rates (HR) in beats/minute (bpm) after 4 days of treatment beginning at 48 h post fertilization (hpf). (**A**) This figure indicates that both AI (50 µM) and general (g) NOSI (15 mM) significantly reduces (*p* < 0.001) HR; (**B**) These data indicate that gNOSI + NO (50 µM) co-treatment significantly rescues (** *p* < 0.05) HR when compared to that of gNOSI; (**C**) On the other hand, gNOSI can prevent the rescue of HR caused by E2 (10 nM) replacement therapy. Specifically, if gNOSI is added to the AI + E2 treatment paradigm, the rescue of HR was significantly eliminated (*p* < 0.001). All bars = ± SD. Asterisks indicate significant differences between treatment groups and controls.

**Figure 2 brainsci-06-00051-f002:**
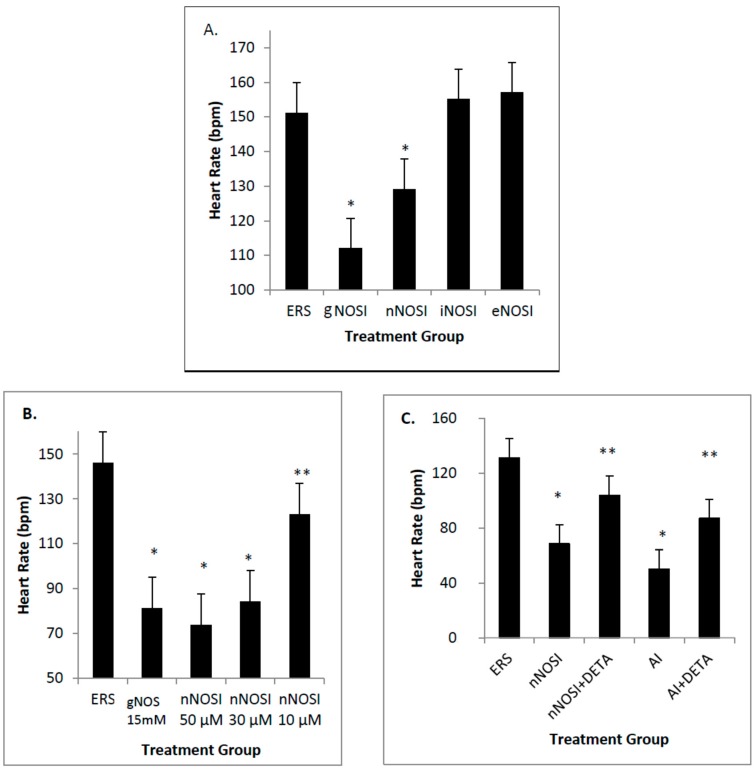
The effects of various treatments on fish heart rates (HR) in beats/minute (bpm) after 4 days of treatment beginning at 48 h post fertilization (hpf). (**A**) The only significant decrease in HR among the three NOS isoform inhibitors was induced by the nNOSI treatment (*p* < 0.001). The nNOSI (50 µM) treatment is shown to closely mimic the significant decrease in HR initiated by the gNOSI (15 mM) treatment; (**B**) This figure shows a dose–response analysis for varying concentrations of nNOSI on fish HRs. ERS (control embryo rearing solution) was used as a control to measure the effectiveness of the different nNOSI concentrations. Specifically, data shows that nNOSI at 30 and 50 μM lowered HR in a dose dependent manner when compared to the 10 µM concentration (*p* < 0.05 for both). Note that gNOSI also significantly reduced HR in a manner similar to that of nNOSI; (**C**) This figure shows the HR rescue effect initiated by combining both nNOSI or AI (50 µM) with DETA-NO (50 µM) as an NO donor (*p* < 0.002 for nNOSI + DETA-NO vs. nNOSI and *p* < 0.001 for AI (50 µM) + DETA vs. AI). Double labeled asterisk group indicate a significant difference from their single asterisk counterparts. All bars = ± SD.

**Figure 3 brainsci-06-00051-f003:**
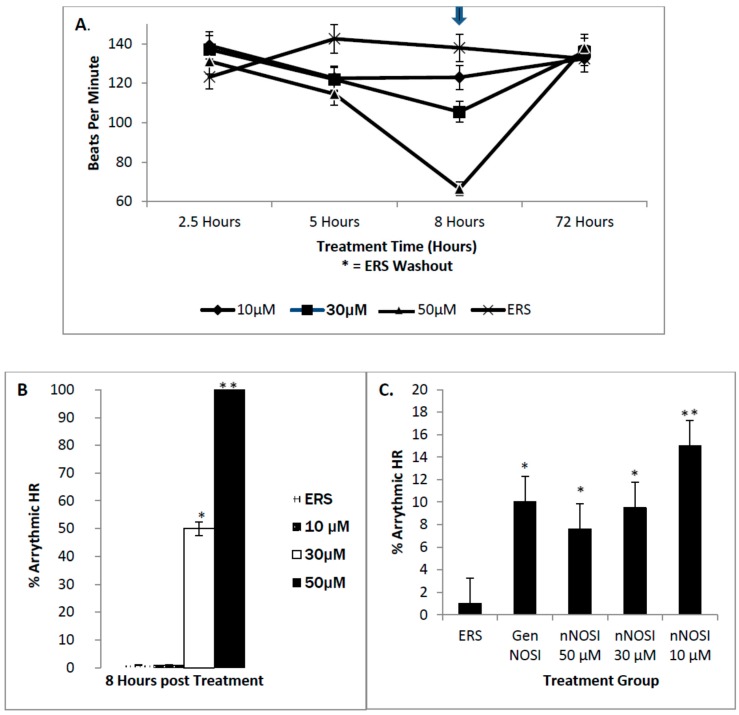
The dose–response effects of nNOSI on zebrafish treated at 2 and 6 dpf and analyzed for heart rate and arrhythmia phenotypes at 4 dpf and 8 h post treatment respectively. (**A**) A depiction of the actual temporal events leading to the significant decrease (* *p* < 0.01) in heart rates (HR) elicited by both 30 and 50 µM nNOSI over the 8 h treatment period in 6 dpf fish. Most significantly, note that 100% of the treated fish populations recover from both the arrhythmic and HR phenotypes when nNOSI is washed out at 8 h of treatment with the ERS control solution (asterisk indicates the time of washout); (**B**) The arrhythmic phenotype in 6 dpf fish can be induced in 100% of the fish population treated with 50 µM nNOSI compared to only 50% affected with a 30 µM concentration both of which are significantly different from each other and that of the ERS controls (* *p* < 0.001). However, no phenotypic expression is induced with a 10 µM treatment which is not significantly different from that of the ERS control fish (*p* < 0.001); (**C**) The effects of different nNOSI concentrations on arrhythmic behavior in fish treated at 2 dpf and analyzed at 4 dpf. Note that the lower concentrations produce the most consistent arrhythmic behavior (*p* < 0.002 for 10 μM and *p* < 0.02 for 30 μM). Also, note that the % arrhythmias in this younger embryonic population (10%–15%) are much less than that seen in the 6 dpf treated fish (compare with Figure 3B above) but are significantly greater (*p* < 0.001) than that of the ERS controls. Bars represent ± SD. Single asterisks represent significant differences between treated groups and controls. Double asterisks indicate a significant difference between dose treatments.

**Figure 4 brainsci-06-00051-f004:**
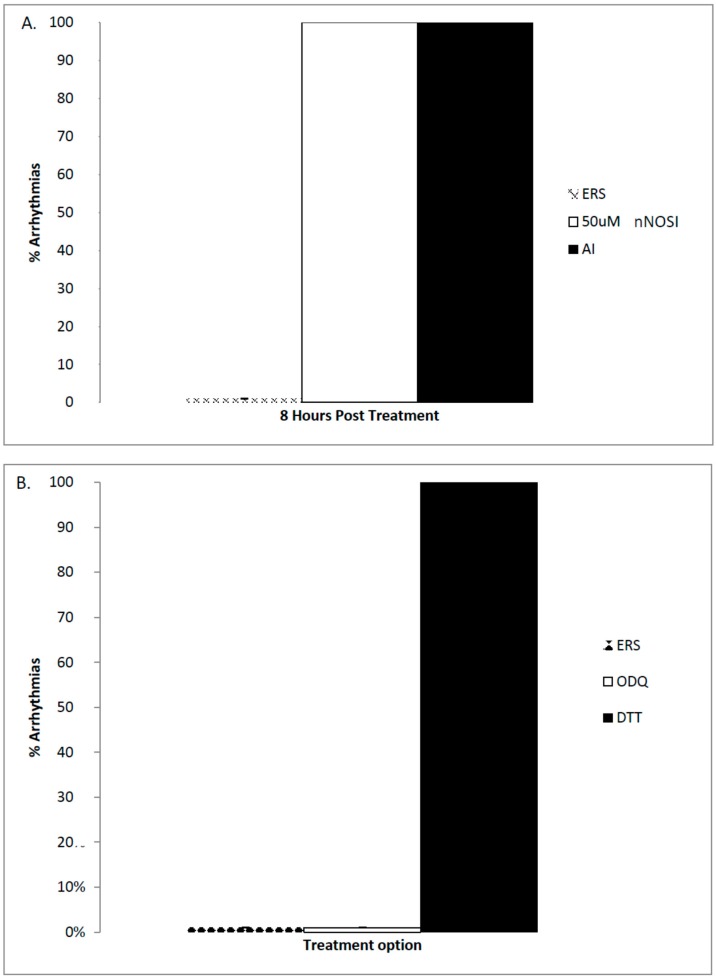
An analysis of arrhythmic behavior under various treatment conditions in the embryonic zebrafish. (**A**) Both AI (50 µM) and nNOSI (50 µM) fish treated at 6 dpf and analyzed 8 h later caused 100% of the population to exhibit the arrhythmic phenotype; (**B**) The demonstration that the arrhythmic phenotype can be attributed to the NO/sGC/cGMP-independent pathway inhibition in fish treated at 6 dpf and analyzed 8 h later. Specifically, fish treated with 100 µM DTT, which reverses S-nitrosylation protein modification, elicits the arrhythmic phenotype in 100% of the fish population. In contrast, no arrhythmic phenotypic expression is evident with a 50 µM ODQ (1H-[1–3]oxadiazolo[4,3-a]quinoxalin-1-one) treatment, which blocks the activity of sGC, and is no different than that of the ERS controls (*p* > 0.05).

**Figure 5 brainsci-06-00051-f005:**
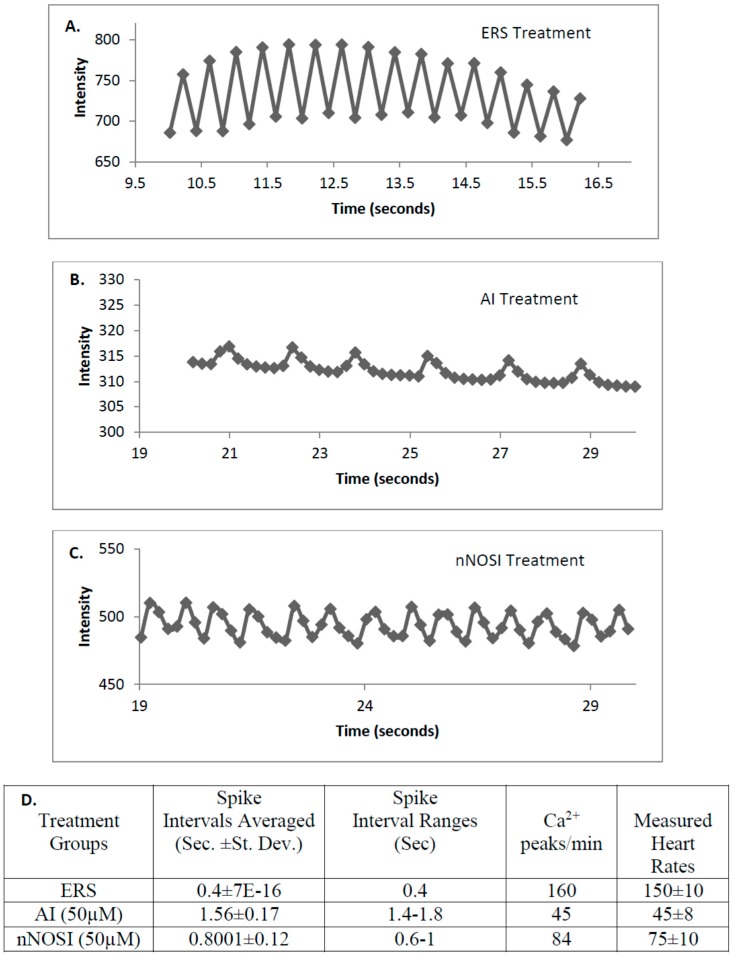
The effects of various treatments on selected representative fish hearts as measured by Ca^2+^ flux patterns in the *Tg(cmlc2:gCaMP)* line of transgenic fish after 4 days of treatment beginning at 48 h post fertilization (hpf). (**A**) This figure shows the Ca^2+^ flux that takes place in a control zebrafish heart treated in ERS solution. Note the regularity of the CA^2+^ flux spike patterns represented by the peaks and valleys. For example, this regular spike interval occurs every 0.4 s (see comparisons in Figure 3D. below); (**B**) This figure shows the calcium flux in a zebrafish heart treated with the aromatase inhibitor (AI) 4-androstene-3,17-dione. Note that peaks are even farther apart than that of nNOSI and ERS controls. The time difference between peaks ranges between 1.4–1.8 s, which are more than four times longer than ERS controls; however, the spike patterns still remain more regular than their nNOSI treated counterparts (see comparisons in Figure 3C,D below); (**C**) This figure shows a more pronounced arrhythmia behavior exhibited by irregular calcium flux patterns in a zebrafish heart treated with 50 μM nNOSI. When compared to ERS controls, the interval differences between peaks were observed to occur over a more extensive and irregular range of 0.6–1 s; (**D**) This table quantifies calcium flux values for the three fish seen in Figure 5A–C above. Specifically, the Ca^2+^ peak values/min for all three fish matches very closely the actual measured heart rates (beats/minute) for comparable treatment groups throughout the study with the AI and nNOSI HRs in the arrhythmic range (<120 bpm, 45 ± 8 and 75 ± 10 bpm respectively, compared to 150 ± 10 for controls) and were further confirmed by their increased and more irregular Ca^2+^ spike intervals.

**Figure 6 brainsci-06-00051-f006:**
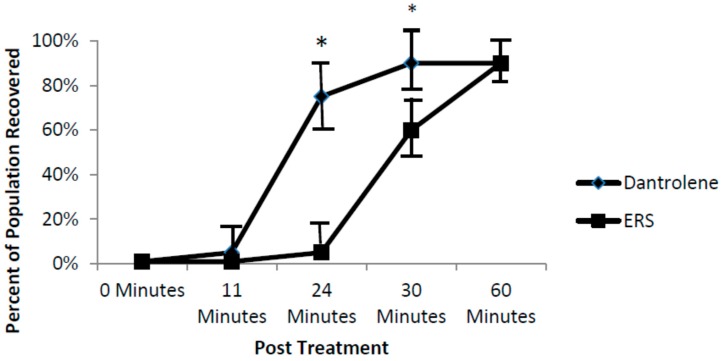
The effects of ERS and dantrolene (10 µM) washout on the recovery of arrhythmic fish previously treated with nNOSI. Note that by 24 min after washout approximately 75% of dantrolene treated fish had recovered from the arrhythmic phenotype while in ERS treated fish recovery was insignificant (*p* < 0.001). By 30 min after ERS washout, although approximately 60% of the fish had recovered from the arrhythmic phenotype, they were still significantly behind the 90% recovery for the dantrolene treated fish (*p* < 0.05). However, by 60 min after washout, both treated fish group HRs had returned to normal heart rate levels. Bars = ± SD. Asterisk indicates significant differences between both 24 and 30 min measures.

**Figure 7 brainsci-06-00051-f007:**
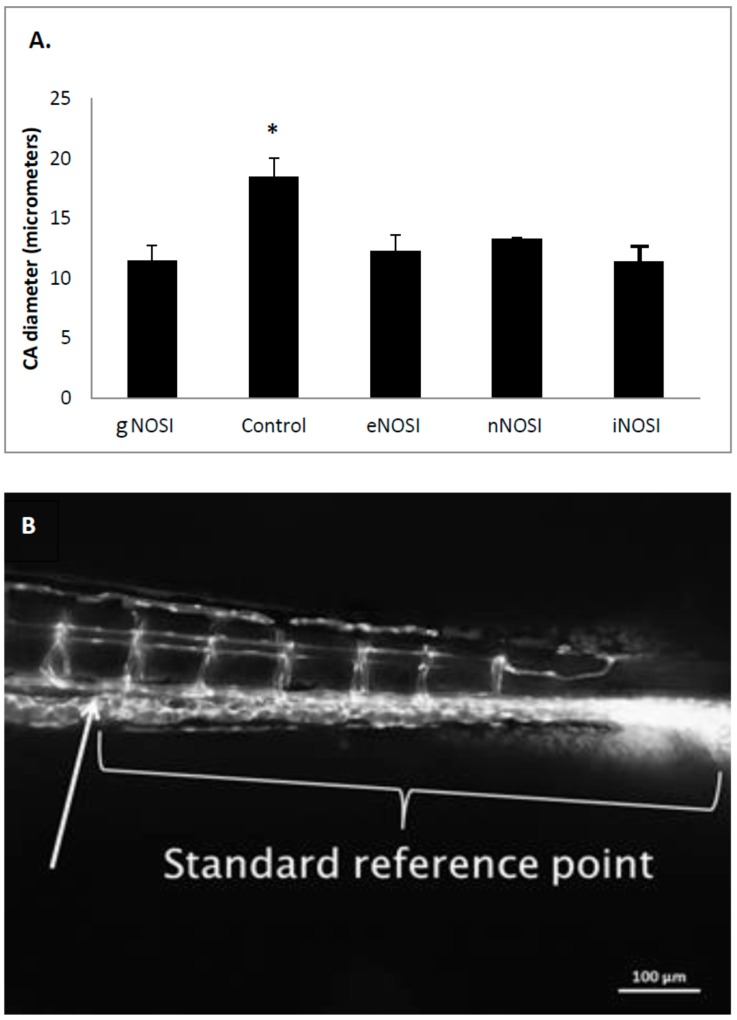
The effects of various treatments in *TG(fli1:EGFP) y1/+y1 (AB)* transgenic fish on average tail caudal artery (CA) diameter as measured from confocal z-stack photomicrographs after 4 days of treatment beginning at 48 h post fertilization (hpf). (**A**) CA diameter is significantly decreased in gNOSI treated fish when compared to that of controls (* *p* < 0.001). In addition, all three NOSI isoforms also cause a significant decrease in CA diameter (*p* < 0.05; Bar = ± SD); (**B**) In order to ensure that measurements of all vessels including the CA were collected in the same location in every confocal- imaged photograph, a standard reference point (arrow) of a distance (875 µm) from the tail region was used. Bar equals 100 µm.

**Figure 8 brainsci-06-00051-f008:**
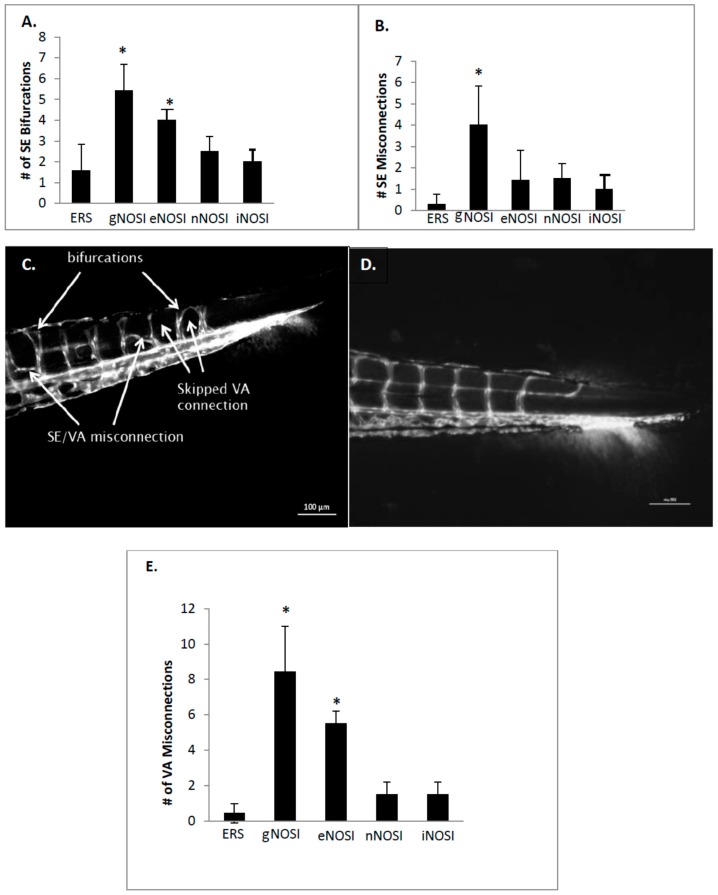
The effects of various treatments in *TG(fli1:EGFP) y1/+y1 (AB)* transgenic fish on average tail blood vessel measurements calculated from confocal z-stack photomicrographs after 4 days of treatment beginning at 48 h post fertilization (hpf). (**A**) This figure indicates that the number of intersegmental vessel (SE) abnormal bifurcations increases significantly when treated with gNOSI (15 mM) or eNOSI (5 µM). Both treatments were significantly more effective than either nNOSI (50 µM) or iNOSI (10 µM) in causing this anomaly (* *p* < 0.001); (**B**) The number of SE misconnections increases significantly when treated with gNOSI compared to the control values (* *p* < 0.001). Also, gNOSI was significantly more effective than eNOSI, nNOSI, and iNOSI in causing this vessel anomaly (* *p* < 0.001); (**C**,**D**) Confocal z-stack imagery of the vasculature shows either retarded growth or deterioration, particularly in the dorsal longitudinal anastomotic vessel (DLAV) and vertebral artery (VA) vessels (arrows) in the various treated groups mentioned in A–B above which appear patchy or absent (arrows) when compared to control fish as seen in D; (**E**) Treatment effects on vertebral artery (VA) development demonstrates significant numbers of misconnections when treated with gNOSI and eNOSI compared to the control values (* *p* < 0.001). Also, note that both eNOSI and gNOSI were significantly more effective than nNOSI and iNOSI in perpetuating this vessel anomaly (* *p* < 0.001). Bars = ± SD. The data indicated in A, B, and D above was collected from the last 875 µM of the caudal area.

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
