# Peer review of "The Relationship between Estrogen and Nitric Oxide in the Prevention of Cardiac and Vascular Anomalies in the Developing Zebrafish (Danio Rerio)"

_brainsci, 2016, doi:10.3390/brainsci6040051_

Round 1

Reviewer 1 Report

Comments on Manuscript BrainSci-152309

Overall comments:  An interesting paper describing the effects of estrogen, nitric oxide inhibitors, and aromatase inhibitors on zebrafish cardiovascular development.  Most of the data and statements are reasonable.  The few problems I have with data presentation will be addressed in comments.

General writing:
1) Many times you use “that” or “which” incorrectly.  If you use “which” a common is required.  No comma is required for “that”
2) A few times you use phrases such as “It has been known that…” or “It has been demonstrated that…”.  These phrases can often be deleted. 

Title:
1) Remove the comma after Cardiac

Abstract:
1) Sometimes you use “a” in front of “NO” and sometimes “an”.  Be consistent.
2) Add a comma after E2 + AI co-treatments
3) A discussion/conclusion sentence or two should be added at the end of the abstract.

Introduction:
1) The intro is particularly long compared to the rest of the paper.  For example, the paragraphs from “There are four isoforms…” to the paragraph “All four isoforms…” could be shortened and combined into a single paragraph that focuses on the NOS isoforms relevant in these experiments.
2) In the second paragraph change “NO has been show” to “NO has been shown”.
3) The sentence on the Nobel Prize is not necessary and should be deleted.
4) In the paragraph beginning with “NO acting”, use a semicolon to avoid a run-on sentence for “subunits; the specific form…”
5) In that same paragraph, “heam” should be “heme”
6) In the third to last paragraph “Zebrafish are commonly…”, you state that nNOS (also known as NOS1) is expressed at 16 to 19 hpf and begins expression at 55 hpf.  Which is it?
7) In that same paragraph, you state that there are many papers that show the importance of nNOS but only have one paper in the reference for that sentence.

Methods:
1) The description of ERS does not include a volume for these solids to be dissolved into.
2) methylene blue is antifungal, not antimicrobial
3) You call the roy;nacre double mutant a transgenic, but these do not contain a transgene.
4) Here and throughout the paper, zebrafish transgenics should be italicized Tg(cmlc2:gCaMP).
5) The first sentence under reagent preparations should have “ensure” instead of “insure”
6) The second paragraph under Reagent Preparations has a reference in the wrong format.

7) In the heart rate measurement paragraph, “160 bmp” should be “160 bpm”

Results:
1) Avoid sentences such as “Figure 1 demonstrates…”.  Instead use sentences such as “Heart rates are affected by nitric oxide and estrogen pathway disruptors (Figure 1).”
2) gNOSI + NO is said to rescue gNOSI, however statistical tests in Figure 1 are between controls and treatments, not between the treatments, according to the figure legend.

Discussion:
1) In the first paragraph, change “Data from the present study suggests” to “Data from the present study suggest”
2) In the first paragraph, add a comma so that the sentence reads “E2 + AI co-treatments, fish could be significantly…”
3) In the second paragraph, delete the comma between “tested” and “nNOS”
4) In the second paragraph, change “Results from the current study again confirms” to “Results from the current study again confirm”

Figures and Figure Legends:
1) In Figure 1, clarify the statistical comparisons.  Asterisks above all samples is not informative.  Are all samples being compared to control or are samples such as “gNOSI + NO” being compared to “gNOS” alone.
2) Add the concentrations being used for each compound to the figure legends in all figures.  Many figures do not have the concentration listed anywhere.
3) In Figure 2, are the “10-5” after some compounds the treatment concentration?  If so, use “uM” in 2A as was done in 2B.
4) Figure 2C has a Y axis with a decimal point while 2A and 2B do not.  Remove that.
5) Figure 3A uses an asterisk for a washout timepoint instead of statistical significance.  Use a different symbol.
6) Figure 3A needs a line for the 50 uM treatment to be consistent
7) In Figures 3B and 3C, the Y axes should be the same label.
8) Figure 4A and 4B should have the same Y axis.  These figures should also be structured with the key at the same location (either to the right or underneath).
9) Figure 5A shows an alternating pattern of high and low intensity.  A curve similar to those seen in Figure 5B and 5C would be helpful to see variation in the spoke intervals.  Label A, B, and C with the treatment. 
10) In Figure 6 legend, should 22 minutes be 24 minutes?  An asterisk is only over 24 minutes, but not 30 minutes; the figure legend seems to indicate that there is a difference at 30 minutes.
11) Figure 7A, are the “10-5” after some compounds the treatment concentration?  If so, use “uM” in 2A as was done in 2B.
12) Figure 7A needs an error bar on iNOSI
13) Figure 8A and 8B need error bars on some samples.
14) Figure 8A, 8B, and 8E all seem to count an aspect of vessel development but these all use different axes labels (8A has “# of”, 8B has nothing, 8E has “Frequency of”).  These should be standardized.
15) Are the data in 8A, 8B, and 8E the number of anatomical features in the 875 microns mentioned in Figure 7?  Please clarify the area being measured in Figure 8.

Author Response

Author’s Responses

Manuscript ID - Turner

Manuscript BrainSci-152309

“The Relationship Between Estrogen and Nitric Oxide in the Prevention of Cardiac, and Vascular Anomalies in the Developing Zebrafish (Danio rerio)”

Referee #1

I.         Referee’s Comments - General writing:

1) Many times you use “that” or “which” incorrectly.  If you use “which” a common is required.  No comma is required for “that”

Author’s Response: Done (see track changes)

2) A few times you use phrases such as “It has been known that…” or “It has been demonstrated that…”.  These phrases can often be deleted. 

Author’s Response: Done (see track changes)

II.       Referee’s Comments - Title:

1) Remove the comma after Cardiac

Author’s Response: Done (see track changes)

III.     Referee’s Comments Abstract:

1) Sometimes you use “a” in front of “NO” and sometimes “an”.  Be consistent.

Author’s Response: Done (see track changes)

2) Add a comma after E2 + AI co-treatments

Author’s Response: Done (see track changes)

3) A discussion/conclusion sentence or two should be added at the end of the abstract.

Author’s Response: Done (see track changes)

IV.    Referee’s Comments - Introduction:

1) The intro is particularly long compared to the rest of the paper.  For example, the paragraphs from “There are four isoforms…” to the paragraph “All four isoforms…” could be shortened and combined into a single paragraph that focuses on the NOS isoforms relevant in these experiments.

Author’s Response: Introduction <40% (see track changes)

2) In the second paragraph change “NO has been show” to “NO has been shown”.

Author’s Response: Done (see track changes)

3) The sentence on the Nobel Prize is not necessary and should be deleted.

Author’s Response: Done (see track changes)

4) In the paragraph beginning with “NO acting”, use a semicolon to avoid a run-on sentence for “subunits; the specific form…”

Author’s Response: Done (see track changes)

5) In that same paragraph, “heam” should be “heme”

Author’s Response: Sentence removed containing “heam” to shorten introduction (see track changes)

6) In the third to last paragraph “Zebrafish are commonly…”, you state that nNOS (also known as NOS1) is expressed at 16 to 19 hpf and begins expression at 55 hpf.  Which is it?

Author’s Response: last part of sentence deleted …”and begins at 55 hpf.” (see track changes)

7) In that same paragraph, you state that there are many papers that show the importance of nNOS but only have one paper in the reference for that sentence.

Author’s Response: Sentence wording changed to read “…Overall, nNOS plays an important role in the zebrafish cardiovascular system [11].

V.      Referee’s Comments - Methods:

1) The description of ERS does not include a volume for these solids to be dissolved into.

Author’s Response: Done (See track changes)

2) methylene blue is antifungal, not antimicrobial

Author’s Response: Done (See track changes)

3) You call the roy;nacre double mutant a transgenic, but these do not contain a transgene.

Author’s Response: “transgenic” deleted (see track changes)

4) Here and throughout the paper, zebrafish transgenics should be italicized Tg(cmlc2:gCaMP).

Author’s Response: Done (See track changes)

5) The first sentence under reagent preparations should have “ensure” instead of “insure”

Author’s Response: Done (See track changes)

6) The second paragraph under Reagent Preparations has a reference in the wrong format.

Author’s Response: Done (See track changes)

7) In the heart rate measurement paragraph, “160 bmp” should be “160 bpm”

Author’s Response: Done (See track changes)

VI.    Referee’s Comments - Results:

1) Avoid sentences such as “Figure 1 demonstrates…”.  Instead use sentences such as “Heart rates are affected by nitric oxide and estrogen pathway disruptors (Figure 1).”

Author’s Response: Done (see track changes)

2) gNOSI + NO is said to rescue gNOSI, however statistical tests in Figure 1 are between controls and treatments, not between the treatments, according to the figure legend.

Author’s Response: Added ** to indicate comparison between gNOSI + NO and gNOSI (see track changes)

VII.   Referee’s Comments - Discussion:

1) In the first paragraph, change “Data from the present study suggests” to “Data from the present study suggest”

Author’s Response: Done (see track changes)

2) In the first paragraph, add a comma so that the sentence reads “E2 + AI co-treatments, fish could be significantly…”

Author’s Response: Done (see track changes)

3) In the second paragraph, delete the comma between “tested” and “nNOS”

Author’s Response: Done (see track changes)

4) In the second paragraph, change “Results from the current study again confirms” to “Results from the current study again confirm”

Author’s Response: Done (see track changes)

VIII. Referee’s Comments - Figures and Figure Legends:

1) In Figure 1, clarify the statistical comparisons.  Asterisks above all samples is not informative.  Are all samples being compared to control or are samples such as “gNOSI + NO” being compared to “gNOS” alone.

Author’s Response: Done. See ** addition comments in VI 2. Above. (see track changes)

2) Add the concentrations being used for each compound to the figure legends in all figures.  Many figures do not have the concentration listed anywhere.

Author’s Response: Done. Where not indicated in figures, all concentrations added to figure legends

3) In Figure 2, are the “10-5” after some compounds the treatment concentration?  If so, use “uM” in 2A as was done in 2B.

Author’s Response: Cone (see track changes)

4) Figure 2C has a Y axis with a decimal point while 2A and 2B do not.  Remove that.

Author’s Response: Done (see track changes)

5) Figure 3A uses an asterisk for a washout timepoint instead of statistical significance.  Use a different symbol.

Author’s Response: Replaced * with arrow (see track changes)

6) Figure 3A needs a line for the 50 uM treatment to be consistent

Author’s Response: Done (see track changes)

7) In Figures 3B and 3C, the Y axes should be the same label.

Author’s Response: Done (see track changes)

8) Figure 4A and 4B should have the same Y axis.  These figures should also be structured with the key at the same location (either to the right or underneath).

Author’s Response: Done (see track changes)

9) Figure 5A shows an alternating pattern of high and low intensity.  A curve similar to those seen in Figure 5B and 5C would be helpful to see variation in the spoke intervals.  Label A, B, and C with the treatment. 

Author’s Response: Figure 5B and C are not curves but the actual spike heights. A,B, and C have been labeled with the treatment (see track changes)

10) In Figure 6 legend, should 22 minutes be 24 minutes?  An asterisk is only over 24 minutes, but not 30 minutes; the figure legend seems to indicate that there is a difference at 30 minutes.

Author’s Response: 22 min. changed to 24 and an * added at 30 min. to show significance (see track changes)

11) Figure 7A, are the “10-5” after some compounds the treatment concentration?  If so, use “uM” in 2A as was done in 2B.

Author’s Response: Done (see track changes)

12) Figure 7A needs an error bar on iNOSI

Author’s Response: Done (see track changes)

13) Figure 8A and 8B need error bars on some samples.

Author’s Response: Done (see track changes)

14) Figure 8A, 8B, and 8E all seem to count an aspect of vessel development but these all use different axes labels (8A has “# of”, 8B has nothing, 8E has “Frequency of”).  These should be standardized.

Author’s Response: Changed all to # symbol (see track changes

15) Are the data in 8A, 8B, and 8E the number of anatomical features in the 875 microns mentioned in Figure 7?  Please clarify the area being measured i

Author’s Response: Yes, explanation added to end of figure legend (see track changes)

Reviewer 2 Report

This manuscript is well written and well executed. The role of Estrogen and NO and their link in cardiovascular field has been known for decades except in Zebra fish. However, the experiments are well designed and conveyed the message as appropriate to the Zebra fish scientific groups.

Introduction need to be shortened. Consider to move most of it to the discussion part and integrate as appropriate. Figure (histograms) presentations need to be improved (all bars were given unfilled). Need to use appropriate symbols in the figure legends. ('u' has been used for micro symbol).

Author Response

Author’s Responses

Manuscript ID - Turner

Manuscript BrainSci-152309

“The Relationship Between Estrogen and Nitric Oxide in the Prevention of Cardiac, and Vascular Anomalies in the Developing Zebrafish (Danio rerio)”

Referee #2

Referee’s Comments #1: Introduction need to be shortened. Consider to move most of it to the discussion part and integrate as appropriate.

Author’s Response: We have shortened the Introduction by 40% and placed a majority of the material in the discussion section (see tract changes).

Referee’s Comments #2: Figure (histograms) presentations need to be improved (all bars were given unfilled).

Author’s Response: For the sake of uniformity all specifically  histogram bars were filled as requested (see track changes for Figures ------- ).

Referee’s Comments #3: Need to use appropriate symbols in the figure legends. ('u' has been used for micro symbol).

Author’s Response: Figure 3 legends changed from u to µ notations (see track changes).
